# Online activity of mosques and Muslims in the Netherlands: A study of Facebook, Instagram, YouTube and Twitter

**Frank van Tubergen**[1,2]*, **Tobias Cinjee**[1], **Anastasia Menshikova**[3], **Joran Veldkamp**[1]

**1** Department of Sociology, Utrecht University, Utrecht, Netherlands, **2** Netherlands Interdisciplinary Demographic Institute, KNAW/University of Groningen, The Hague, Netherlands, **3** The Institute for Analytical Sociology, Linköping University, Linköping, Sweden

* f.vantubergen@uu.nl

## Abstract

Research on Muslim minorities in western societies has mainly focused on offline behavior, such as mosque attendance, whereas little is known about their presence in the online world. This study explores the online visibility and activities of all (478) mosques in the Netherlands. We collected data on personal websites and four social media platforms (Facebook, Twitter, Instagram and YouTube). The majority of mosques have a website (52%) and an account on Facebook (61%). Less often used are Twitter (17%), Instagram (17%) and YouTube (19%). On social media platforms, mosques strongly differ in their activity and number of followers. We find evidence to suggest that Salafist mosques, which tend to have a strict ideology, are more active on Twitter and YouTube, and also attract a larger share of followers on Facebook than non-Salafist mosques. Our more fine-grained analysis on Twitter shows that Salafist mosques in the Netherlands cluster together. Followers of Salafist mosques make up a community of users who are mainly connected to each other ("bonding ties"), and much less so to other users ("bridging ties"). We conclude with a discussion of opportunities for studying the online presence and activities of mosques and Muslims in western societies.

## 1. Introduction

The increasing ethnic diversity in Western European societies in the past decades has put questions on religion and immigrant incorporation at the forefront of public and scientific debate. Scholars have noted that a sizeable share of immigrants and their children are devout Muslims [1] and that religion may be a barrier to integration in European host societies which are predominantly secular-Christian [2, 3]. Studies report that in Western Europe, ethnic majority members have strong negative attitudes toward Muslims, more so than toward other groups [4] and that Muslims are discriminated in the labor market [5, 6]. Studies on friendship ties in schools show that personal networks are segregated between Muslims and non-Muslim youth [7]. Muslim minority members in Western Europe endorse more conservative norms and values than ethnic majority members [8], and, on average, tend to have more negative views of

**Data Availability Statement:** Data are available here: https://doi.org/10.17026/dans-zem-9g7g.

**Funding:** FvT received support for this research from the Royal Netherlands Academy of Arts and

Sciences (Ammodo KNAW award 2015). The
funder had no role in the study design, data
collection and analysis, decision to publish, or
preparation of the manuscript.

**Competing interests:** The authors have declared
that no competing interests exist.

ethnic majority members and out-groups than toward their own group [9, 10]. Furthermore, scholars have seen an increase in attacks on mosques in Western Europe [11].

Given the significance of Muslim identity in the integration of a large share of ethnic minorities in Western Europe, scholars have examined the religious trajectories of Muslim minority groups [12]. It has been found that in Muslim minority families, there is strong intergenerational transmission of religious identity and religiosity from parents to children [13–15]. Several studies report a slight overall tendency towards secularization across generations among Muslims in Western Europe, as Muslim children appear less religious than their parents on average [16–21]. However, studies have also reported that a small group of Muslim youth has become more religious over time [22], that some even turned to more extremist, radicalized views of Islam [23], and that Salafism, the more orthodox version of Islam, may have increased in popularity among Muslims in Western countries [24].

Most research on Muslims in Western Europe so far has been largely concerned with the *offline* religious behavior of Muslim minorities, such as their religious attendance and frequency of praying. Less is known, however, about *online* Muslim communities, and the connection between online and offline. This is remarkable, because the world has become digitally connected via social media platforms, websites and mobile devices. Studies in the field of religion and the internet, described as 'digital religion' research [25–27], reveal that online Muslim communities can play a key role for Muslims. For example, one study, using data on religiously diverse Singapore, found that many Muslims use the internet for faith-related matters, such as for personal religious concerns, to read about Islam, to search for mosques near their home or to check for prayer times [28]. Another study, using data on the Muslim-Italian blog *Yalla*, describes how Muslims in Italy use *Yalla* as an online platform to discuss their position as Muslims in Italy [29]. Other scholars have looked at extremism and provided evidence to suggest that exposure to jihadist extremist ideology on websites, and social media platforms such as Facebook and YouTube, may create opportunities for Muslims in Western Europe to become radicalized and also act as an 'echo chamber' for extremist beliefs [30].

What is missing from this literature is large-scale comparative research on Muslims' adoption of online platforms. In a recent review of digital religion research, it was concluded that "there needs to be much more research on religious use of Facebook, Twitter, YouTube, Instagram (. . .), on religious groups' adoption of online contexts" [25]. In response to this, and in light of the significance of Muslim identity in Western Europe, we examine the online activities of mosques and their followers in the Netherlands. While previous studies in this field have used a case-study approach, looking, for example, at a single Muslim website, a specific Islamic Facebook page or a single Muslim minority group, we analyze the presence of all mosques and their followers from all Muslim minority groups in the Netherlands on websites, Facebook, Twitter, YouTube and Instagram.

Taking such a large-scale quantitative comparative approach contributes to previous work in three ways.

First, it allows us to examine differences between Muslim minority groups in adopting online contexts. Case studies offer rich descriptive detail, but findings are difficult to generalize to other Muslim groups. Because we include all Muslim groups, we are able to study which groups are more active online, and to test hypotheses that explain such group differences. We study whether organizational resources or ideology serve as better predictors for online activity.

Second, we are able to compare adoption rates across online platforms. Previous studies have focused on one online platform at a time (e.g., Facebook), but Muslim groups can have differential activity rates across different online platforms (e.g., Facebook vis-à-vis Twitter). Previous studies found that social categories (e.g., age, gender) are unevenly represented on

different social media platforms such as Twitter, Instagram and Facebook [31], and we extend this type of research by examining differences between Muslim groups in adopting websites, Facebook, Twitter, YouTube and Instagram.

A third contribution of our comparative quantitative study is that we look at the cohesiveness of Muslim communities. A major gap in the literature is that little is known about the community formation of Muslim groups online, e.g., whether some Muslim groups have a dense, cohesive cluster of followers, whereas other groups are more fragmented in the online world. Cohesiveness, and network closure, are key concepts in understanding how 'closed' or 'open' groups are to members of other groups, to other views, norms and values [32, 33]. To examine cohesiveness, we study how strongly the connections are between followers of the same mosque in the online world, and whether these followers cluster together in areas nearby the mosque.

The aim of this study is to answer the following research question:

*To what extent and why are there differences between Muslim groups and their followers (a) in the adoption of online contexts (i.e., websites, Facebook, Twitter, YouTube and Instagram), and (b) the cohesiveness of these online communities?*

The context of this study is the Netherlands, a country which used to be predominantly Protestant and Catholic. Because of secularization processes that took place since the 1960s, it has lost its strong Christian identity, and nowadays the majority of the population consider themselves not affiliated to a religion [34]. Parallel to this process, the Netherlands has witnessed a rising share of the population with a migration background. Many of those newcomers and their children are devout Muslims, with the two largest Muslim groups coming from Turkey and Morocco [35]. Immigrants from these two countries arrived at the same time, in the 1960s as part of the 'guest-worker' program of the Dutch government. The group consisted of male, low-skilled workers, and they were followed by their families, who migrated to the Netherlands in the 1970s and 1980s [36]. Today, these Muslim minority groups are roughly equal in size and a large group of their members consist of second or even third generation [37, 38]. Around 95% of those having a Turkish or Moroccan origin in the Netherlands identify themselves as Muslim [18]. Next to these two large groups, there are many smaller Muslim groups in the Netherlands, such as those coming from Somalia, Afghanistan, Iran and Iraq. The case of the Netherlands is quite similar to that of other Western European countries, such as Germany, France and the UK, which have also experienced on the one hand a process of secularization among the (ethnic) majority population and on the other hand a growing proportion of Muslim minorities [1].

## 2. Theory and hypotheses

### 2.1 Adoption of online contexts

Muslim minority groups in Western European countries, including the Netherlands, are exposed to the secular-Christian culture of the (ethnic) majority population [1]. Relying on theories of social control and normative social influence, scholars have argued that Muslim minority groups experience pressure to conform to the mainstream norms and values in various settings, such as in school, at work, and in the neighborhood [35, 39]. As a response to these secularizing-assimilationist forces, Muslim minority groups can build an organizational structure that protects their Muslim identity, and which creates opportunities for religious gatherings and meeting co-religious peers. This organizational layer consists of mosques and Islamic-based voluntary organizations, schools, and shops.

With the rise of the internet and digital communication, religious organizations can also manifest themselves online, such as on Facebook, Instagram, YouTube and Twitter [40]. In the secularized context of Western Europe, mosques can make use of websites and social media platforms, to disseminate Islamic ideology, to post information on praying schedules, to share videos and to create an online community to maintain the Muslim identity of their members.

However, the degree to which mosques adopt online contexts may differ between Muslim minority groups. In this study, we rely on theories of social movements and organizations [41], and examine the role of organizational resources and ideology in explaining group differences.

Drawing on ideas of organizational resource theory [42], we argue that the online community that a Muslim group can build depends on its offline organizational structure. It takes time and effort for organizations to actively maintain websites, to post messages on Facebook and Instagram, and to create and upload interesting videos on YouTube. This means that those Muslim groups that are better organized offline, have more resources to become actively engaged on online social platforms. In addition, when Muslim groups are more organized institutionally, it also indicates that their members are already more embedded in their organization. Thus, there is also a larger online demand and market of followers for Muslim groups that are more institutionalized offline.

To test this theoretical idea, we compare the online activities of two Muslim groups that differ in their (offline) organizational structure. Although there are many Muslim groups in the Netherlands, findings on organizational activities are available only for the two largest Muslim groups, i.e., Turks and Moroccans. These groups, however, provide an interesting comparison because they differ in terms of organizational resources. Studies show that the Turkish community in the Netherlands is much more organized institutionally than the Moroccan group [43]. There are relatively more Turkish than Moroccan organizations, such as mosques and sports clubs. Turkish minorities in the Netherlands are more often member of Turkish organizations compared with Moroccans affiliated to Moroccan organizations.

Based on resource-arguments, this implies that Turkish mosques may be more active online, and that members of Turkish mosques would be more inclined to follow these mosques online. That does not mean that every 'follower' of a mosque online also participates in meetings of that mosque offline. Everyone can follow mosques online, and thus a 'follower' of a Turkish mosque on Twitter is not a person per se who attends meetings of that mosque. However, this applies to all mosques we study, and ceteris paribus, the organizational resource argument leads to the following testable hypothesis:

H1. In the Netherlands, Turkish mosques and their followers are more active on online platforms than Moroccan mosques and their followers.

The second theoretical mechanism that we examine is related to ideology. The differential adoption of online contexts can be due to varying religious strictness of Muslim groups. In the sociology of religion literature, scholars have argued that some religious groups are more 'strict' or 'orthodox' than other, more 'lenient' religious groups. Strict religious communities place greater penalties to free riding, screening out members who lack commitment [44]. According to this idea, strict religious organizations demand more from their members than more lenient organizations. This includes, amongst others, that members from strict groups are expected to devote their time and energy to the benefits of the group rather than to non-group activities. This may give strict religious groups a competitive advantage, because their members are strongly participating in religious meetings, they are actively engaged in recruiting new members, and strongly prohibit members leaving their community. The empirical

evidence for this "strictness hypothesis" is a topic of discussion in the sociology of religion [45, 46].

We use this theoretical mechanism and test the idea that Muslim groups with a strict ideology are more active online than more lenient Muslim groups. In the context of Islam, an often-made distinction in terms of strictness is between Salafist and non-Salafist groups. Salafism is a movement, which, generally speaking, stands for a very strict, conservative interpretation within Islam which goes back to the early years of the religion [24, 47]. Salafist groups demand from their followers that they are strongly committed to promote Salafism ideology, that they are actively engaged and recruit new members. From the ideology-argument, we derive the following testable hypothesis:

H2. In the Netherlands, Salafist mosques and their followers are more active on online platforms than non-Salafist mosques and their followers.

## 2.2 Cohesiveness

We argue that the ideology of Muslim groups can also impact their online cohesiveness. The degree of 'cohesiveness', also called 'network closure' or 'community', is a key concept in research on networks and groups [48, 49]. It indicates the extent to which members of the same group have ties with each other ('group-bonding') rather than with out-group members ('group-bridging'). Cohesive groups can strongly monitor and sanction free-rider behavior and enforce norms [32]. Scholars have argued that immigrant groups that are more cohesive, are better able to transmit the minority language, religion, cultural values and norms from one generation to the next [50, 51].

Empirically, the cohesiveness of immigrant groups has been studied in various ways. Commonly used measures are endogamy patterns [52], and friendship networks of adolescents in schools [53]. Findings for the Netherlands reveal that the Muslim immigrant groups have higher endogamy rates than non-Muslim immigrant groups [54], and that the friendship networks of Muslim youth are more closed compared to other groups [55]. These studies, however, show that the two largest Muslim minority groups in the Netherlands, i.e., Turks and Moroccans, have very similar levels of cohesiveness. We therefore do not expect to see any difference in the cohesiveness of Turkish and Moroccan mosques and their followers on online platforms.

We do, however, argue that religious orthodoxy plays a role in online group closure. According to Salafist ideology, members are expected, amongst others, to wear traditional clothes, to preserve traditional gender roles, and to refrain from social interactions with ethnic majority members [47]. Because of these self-imposed restrictions for interactions with out-group members, and the cultural distance with the secular Dutch majority population, we argue that both Salafist and Dutch majority members have few incentives to socially connect with each other in the online world. A possible implication, which we study here, is that the online followers of Salafist mosques make up a more cohesive group than those following non-Salafist mosques. Based on the ideology-argument, we hypothesize that:

H3. In the Netherlands, Salafist mosques and their followers are more cohesive groups on online platforms than non-Salafist mosques and their followers.

## 3. Data and measures

This research was approved by the Board of the Ethics Committee of the Faculty of Social and Behavioural Sciences, Utrecht University (approval number 20–339). Consent was not

obtained, because the study uses large-scale, publicly available online data. All data were analyzed anonymously. The population of interest in our study are all mosques in the Netherlands. Unfortunately, there does not exist an official list of all mosques in the Netherlands. We therefore compiled our own list of mosques, identifying 478 in total. Of those, 432 were found on a well-known website which documents the location of mosques in the Netherlands [56]. Of the remaining 46 mosques, 39 were found via a variety of published sources [24, 57–65] and 7 mosques were identified via search on Google, Facebook, Twitter, Instagram and YouTube, and via exploratory social network analyses of known mosques on Facebook and Twitter, which resulted in discovering yet unidentified mosques.

The data collection took place between 15[th] September 2018 and 1[st] July 2019. During this period, we collected data on the online activity of mosques in the Netherlands as well as data on their online followers. We categorized the *ethnic group* to which each mosque belonged, i.e., the dominant ethnic-national orientation of a mosque (e.g., Turks, Moroccan). To do so, we relied on information available from the mosques' websites and their pages on social media platforms. Signals for ethnic group affiliation include language usage (e.g., Arabic, Turkish), presentation of the national flag, or explicit mentioning of the ethnic group to which the mosque belongs. We also relied on an existing source which classified the ethnic affiliation of mosques in the Netherlands [56]. Out of the 478 mosques, we identified 222 as Turkish, and 170 as Moroccan. The remaining 86 mosques refer to (1) a number of smaller ethnic groups (e.g., Surinamese, Indonesian, Pakistani), (2) to mosques having a 'generic' profile (i.e., no specific ethnic group), and to (3) 19 mosques for which we could not identify their ethnic group membership.

Our list of Salafist mosques was based on existing literature in the Netherlands, which identified 23 mosques (5%) as having a Salafist orientation at the time of our data collection [24, 47, 66, 67]. We therefore relied on prior research that coded mosques as Salafist based on a variety of sources, such as Salafist self-identification of mosques, anthropological fieldwork, and information on the ideology of mosques. Although there may be, obviously, varying degrees of strictness between Salafist mosques, as well as between non-Salafist mosques, we were not able to construct such more refined measures of strictness. Importantly, there is an overlap between the ethnic group to which mosques belong and Salafist orientation, with Salafist mosques being overrepresented in the Moroccan group. Specifically, out of the 23 mosques, 14 are identified as Moroccan. There are no Turkish mosques with a Salafist orientation. For this reason, we present our findings in two ways. First, we present tables with the gross differences between Turks and Moroccans, and between Salafist and non-Salafist mosques (main text). Second, we include tables with results from multivariate regression analyses in the Supporting Information, see S1 Text, S1–S5 Tables and we discuss these findings in the main text as well. Detailed information about methodology and measurement can be found in the Supporting Information as well, See S2 Text.

## 4. Results

### 4.1. Mosques: Online presence and activity

We first examined data on the online presence of mosques, as indicated by having a website and/or an account on a social media platform. We checked the social media pages extensively, one by one, and included pages on mosques in the Netherlands that are created by (a) an organizational account of the mosque, or (b) a person on behalf of the mosque. To qualify for (b), (a) should not be present on the same social media platform, and, moreover, it should be clear from the content and number of followers that it is in fact the main social media account (e.g., on Facebook) that is used to spread information about the mosque.

**Table 1. Online presence of mosques in the Netherlands, by ethnic group and strictness (N = 478).**

| | Mosques (N) | Website (%) | Facebook (%) | Twitter (%) | Instagram (%) | YouTube (%) | Average presence (0–5) |
|---|---|---|---|---|---|---|---|
| *All mosques* | 478 | 52.3 | 60.7 | 16.9 | 17.2 | 18.6 | 1.66 |
| *Ethnic group* | | | | | | | |
| Turkey | 222 | 45.0 | 68.0 | 11.7 | 27.5 | 11.7 | 1.64 |
| Morocco | 170 | 55.3 | 54.7 | 20.0 | 6.5 | 25.9 | 1.62 |
| Surinam | 37 | 70.3 | 56.8 | 13.5 | 2.7 | 24.3 | 1.68 |
| Indonesia | 7 | 71.4 | 14.3 | 14.3 | 42.9 | 14.3 | 1.57 |
| Pakistan | 5 | 100.0 | 80.0 | 40.0 | 0.0 | 60.0 | 2.80 |
| Afghanistan | 3 | 67.7 | 67.7 | 0.0 | 0.0 | 0.0 | 1.33 |
| Somalia | 2 | 50.0 | 50.0 | 50.0 | 0.0 | 0.0 | 1.50 |
| India | 1 | 100.0 | 0.0 | 100 | 0.0 | 0.0 | 2.00 |
| Bosnia-Herzegovina | 1 | 0.0 | 0.0 | 0.0 | 0.0 | 0.0 | 0.00 |
| Arabic (general) | 5 | 60.0 | 80.0 | 60.0 | 0.0 | 0.0 | 2.00 |
| Multiple | 6 | 50.0 | 50.0 | 33.3 | 16.7 | 16.7 | 1.67 |
| Unknown | 19 | 47.4 | 47.4 | 31.6 | 26.3 | 26.3 | 1.79 |
| *Strictness* | | | | | | | |
| Salafist | 23 | 73.9 | 60.9 | 56.5 | 4.3 | 21.7 | 2.17 |
| Non-Salafist | 455 | 51.2 | 60.7 | 14.9 | 17.8 | 18.5 | 1.63 |

The majority of mosques have a website (52%) and an account on Facebook (61%). Less often used are Twitter (17%), Instagram (17%) and YouTube (19%). We inspected whether the online presence of mosques varies by ethnic group and strictness (Table 1). Results suggest substantial heterogeneity among ethnic groups in their online presence, and, moreover, in the specific online outlet they manifest themselves. Pakistani mosques in the Netherlands are strongly present online: all five have a website, 80% have a Facebook page, 40% are on Twitter and 60% on YouTube. On average, Pakistani mosques in the Netherlands are visible on 2.8 out of five online outlets (i.e., website, Facebook, Twitter, Instagram and YouTube).

We used the R package 'stats' to perform regression analyses of the overall presence (range 0–5) and of the presence on each platform separately (0/1) (S1 Table). The results suggest that there are no statistically significant differences between Turkish and Moroccan mosques in their overall online presence, i.e., the average of the five online outlets. When comparing the two groups for each online outlet, it appears that Moroccan mosques are significantly less often on Facebook (OR = 0.55, $p$ = 0.006; S1 Table), less often using Instagram (OR = 0.19; $p < 0.001$; S1 Table), and more often present on YouTube (OR = 2.67; $p < 0.001$; S1 Table). There are no significant differences between Moroccans and Turks regarding website usage (OR = 1.43; $p$ = 0.087; S1 Table). Overall, these findings provide no evidence to support H1.

Salafist mosques are present on 2.17 out of five online outlets, compared to 1.63 among non-Salafist mosques, the difference being non-significant at conventional levels ($p$ = .068; S1 Table). Among the 23 Salafist mosques, 74% have a website, whereas this is so for 51% among the non-Salafist mosques ($p$ = .136; S1 Table). Salafist mosques are more active on Twitter than non-Salafist mosques: 57% of the Salafist mosques are active on this platform, while this is true for only 15% of the non-Salafist mosques ($p < .001$; S1 Table). There is therefore some support for H2.

Having a website and an account on social media platforms like Facebook and Instagram obviously does not imply that mosques are actively using these. Indeed, closer inspection of the mosque websites reveals that most provide basic information, such as name and address of the mosque, and contact information. Websites are not often used to actively disseminate

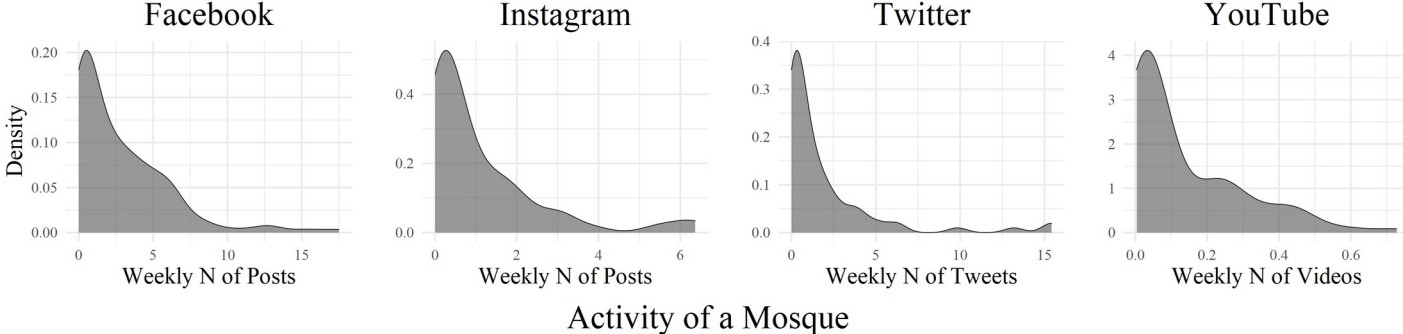

**Fig 1. Distribution of mosque activity on four social media platforms in the Netherlands.** *Note*: Figures include only those mosques that are present on the respective social media platforms.

news and mosque-related activities. Instead, such activities are more often posted on their social media pages.

The extent to which mosques are actively using social media differs, however. We constructed measures of mosque activity on each of the four social media platforms, among those mosques that are present on these platforms. Specifically, we measured weekly averages regarding posts (Facebook, Instagram), tweets (Twitter), and videos (YouTube). We find that the distribution of activity is highly skewed (Fig 1). Among those mosques having an account on Facebook, for example, we find that 90% of the mosques upload fewer than 6.36 posts per week on Facebook, while the two most active mosques upload 17.5 posts on average per week.

Examining averages per platform (Table 2), we find that mosques are most actively using Facebook, with 2.8 posts on average per week. Twitter comes second (1.9 tweets), followed by Instagram (1.1 posts) and then YouTube (0.14 videos per week). Turkish mosques use Facebook more frequently than Moroccan mosques. Specifically, Turkish mosques posted on average 3.27 messages per week on Facebook, compared to 2.31 among Moroccan mosques; a statistically significant difference ($p = 0.029$; S2 Table). Regarding other social media platforms, we do not find any difference. Salafist mosques appear to upload YouTube videos more often than non-Salafist mosques ($p = 0.022$; S2 Table), but with respect to other social platforms, there are no statistically significant differences. Overall, these results speak against H1, while there is some support for H2.

**Table 2. Activity of mosques on social media platforms in the Netherlands, by ethnic group and strictness.**

|  | Facebook (average posts per week) | | Twitter (average tweets per week) | | Instagram (average posts per week) | | YouTube (average videos posted per week) | |
|---|---|---|---|---|---|---|---|---|
|  | Mean | SD | Mean | SD | Mean | SD | Mean | SD |
| *All mosques* | 2.81 | 3.24 | 1.93 | 2.23 | 1.13 | 1.50 | 0.14 | 1.50 |
| *Ethnic group* |  |  |  |  |  |  |  |  |
| Turkey | 3.27 | 3.68 | 1.53 | 2.18 | 1.28 | 1.49 | 0.08 | 0.14 |
| Morocco | 2.31 | 2.23 | 1.31 | 1.32 | 0.29 | 0.49 | 0.16 | 0.17 |
| *Strictness* |  |  |  |  |  |  |  |  |
| Salafist | 2.34 | 2.31 | 3.45 | 3.73 |  |  |  |  |
| Non-Salafist | 2.84 | 3.28 | 1.66 | 3.09 |  |  |  |  |

*Note*: activity measures calculated for only those mosques that are present on respective social media platform. With respect to Instagram and YouTube, numbers for Salafist mosques are too small for descriptive comparisons.

### 4.2 Followers: Numbers and activity

How many people are following the online updates of mosques? It appears that the number of followers strongly differs across mosques (Fig 2). The distribution is highly skewed, with a majority being followed by a small group, and a few mosques attracting many followers. To illustrate, 90% of the mosques with a Facebook account have less than 3000 followers, with an average of 840. The mosque with the highest number of adherents on Facebook has 9798 followers. Among the mosques on Instagram, 90% have fewer than 900 followers (and on average: 258), while the most popular mosque has 2014 subscribers. On Twitter and YouTube, the distribution is even more skewed. About 90% of all mosques have less than 550 followers on Twitter (and on average: 99), while the most popular account has 4968 followers. The second most popular one has 1309 followers. On YouTube, 90% of the mosques have 600 or less followers (and on average: 67), whereas the two most popular mosques have 4454 and 4357 followers.

Looking at averages, we find that mosques are most popular on Facebook, with an average of 1237 followers, followed by Instagram with an average of 361 followers (Table 3). Contrary to H1, there are no statistically significant differences between Turkish and Moroccan mosques regarding their popularity on the four social media platforms. In line with H2, we do find that Salafist mosques attract much more followers online than non-Salafist mosques. The Salafist mosques that have a Facebook page are followed by 2435 people on average, significantly more than non-Salafist mosques that have an average of 1169 ($p$ = 0.012; S3 Table).

Additional analysis reveals a statistically significant positive relationship between the activity of a mosque on a certain platform and the number of followers that mosque attracts on that platform. The degree of association varies across platforms, however. We find a weak correlation for Facebook ($r$ = 0.18), moderately strong correlations for Instagram ($r$ = 0.39) and YouTube ($r$ = 0.38), and a strong correlation for Twitter ($r$ = 0.76). Thus, when mosques more actively maintain their social media pages, they tend to be more popular, and this especially true for Twitter.

Following a mosque on Twitter or Facebook does not say much, of course, about how actively those people are engaged with this mosque. Therefore, in an attempt to get more insight into this, we constructed measures capturing the activity of the online mosque-followers. We find that the distribution of user activity is highly skewed: a few mosques have highly active users, but most less so (S1 Fig). Looking at averages (Table 4), we see that followers of Turkish mosques are significantly more actively viewing YouTube videos from their mosque than followers of Moroccan mosques ($p$ = 0.000; S4 Table). Subscribers of Moroccan mosques, however, are more active on Instagram than are followers of Turkish mosques ($p$ = 0.000;

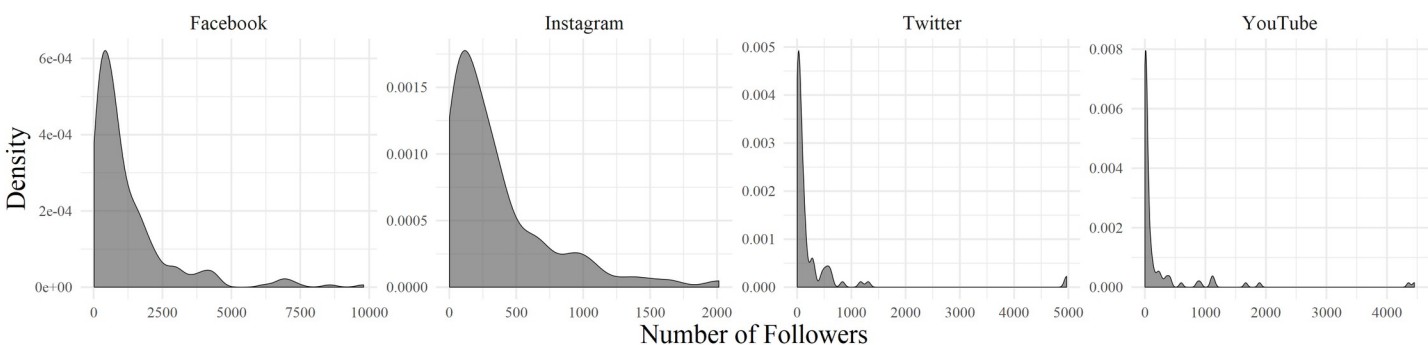

**Fig 2. Distribution of followers of mosques on four social media platforms in the Netherlands.**

**Table 3. Number of followers of mosques on social media platforms in the Netherlands, by ethnic group and strictness.**

|  | Facebook | | Twitter | | Instagram | | YouTube | |
|---|---|---|---|---|---|---|---|---|
|  | Mean | SD | Mean | SD | Mean | SD | Mean | SD |
| *All mosques* | 1237.55 | 1497.71 | 275.60 | 791.74 | 361.61 | 401.20 | 259.88 | 727.30 |
| *Ethnic group* | | | | | | | | |
| Turkey | 946.23 | 1111.56 | 70.65 | 101.78 | 398.10 | 365.49 | 17.96 | 49.10 |
| Morocco | 1430.32 | 1633.04 | 190.65 | 258.99 | 161.64 | 321.97 | 338.00 | 759.86 |
| *Strictness* | | | | | | | | |
| Salafist | 2435.29 | 2357.32 | 468.15 | 447.09 | | | | |
| Non-Salafist | 1169.39 | 1410.61 | 238.79 | 839.11 | | | | |

*Note*: popularity measures calculated for only those mosques that are present on respective social media platform. With respect to Instagram and YouTube, numbers for Salafist mosques are too small for comparisons.

S4 Table). We do not find significant differences between followers of Salafist and non-Salafist mosques in their activity. Overall, these findings are against H1 and H2.

## 4.3. Followers: Cohesiveness

To test the hypothesis on cohesiveness (H3), we study in more detail mosques and their followers on Twitter. This platform was chosen for further analysis, for various reasons. One reason is that Facebook restricted access to researchers at the time of data collection, not allowing researchers to study pages of followers, and thereby to identify their geolocation and their connections to other followers. Furthermore, the number of Salafist mosques on Instagram and YouTube was too small for more-refined analysis.

We were able to analyze 78 Twitter accounts of mosques in the Netherlands. This resulted in 11943 unique followers, of which 10152 could be included in the analysis due to the restrictions we posed on accessibility via the Twitter API (e.g., public profile, profile description). For data collection, we used the R package 'rtweet'. We measured cohesiveness in different ways, namely by looking at geographical clustering of the people who follow the same mosque on Twitter, and by the degree to which followers of the same mosque are connected to each other on Twitter. We use the results from these different measures to evaluate the empirical evidence for H3.

As a first measure of the cohesiveness of Muslim communities, we looked at patterns of geographical embeddedness of the people who follow the same mosque on Twitter. When

**Table 4. Activity of followers from mosques on social media platforms in the Netherlands, by ethnic group and strictness.**

|  | Facebook (N of comments/N of followers) | | Twitter (N of likes/N of followers) | | Instagram (N of likes/N of followers) | | YouTube (N of views/N of followers) | |
|---|---|---|---|---|---|---|---|---|
|  | Mean | SD | Mean | SD | Mean | SD | Mean | SD |
| *All mosques* | 0.44 | 2.44 | 0.01 | 0.03 | 0.16 | 0.27 | 385.28 | 703.45 |
| *Ethnic group* | | | | | | | | |
| Turkey | 0.39 | 2.17 | 0.00 | 0.00 | 0.09 | 0.07 | 856.46 | 1183.38 |
| Morocco | 0.22 | 1.23 | 0.01 | 0.03 | 0.39 | 0.59 | 181.08 | 116.91 |
| *Strictness* | | | | | | | | |
| Salafist | 0.14 | 0.30 | 0.00 | 0.00 | | | | |
| Non-Salafist | 0.45 | 2.50 | 0.01 | 0.03 | | | | |

*Note*: user activity measures calculated for only those mosques that are present on respective social media platform. With respect to Instagram and YouTube, numbers for Salafist mosques are too small for comparisons.

**Table 5. Geographic distance between the mosque and the place of living of their followers on Twitter.**

| | Living in the Netherlands (%) | Among those living in the Netherlands | |
|---|---|---|---|
| | | Average distance to mosque | ≤ 5-kilometer distance to mosque (%) |
| *All mosques* | 80.1 | 40.56 | 28.11 |
| *Ethnic group* | | | |
| Turkey | 81.8 | 24.99 | 46.95 |
| Moroccan | 86.7 | 35.49 | 24.92 |
| *Strictness* | | | |
| Salafist | 82.2 | 38.39 | 22.56 |
| Non-Salafist | 79.7 | 41.33 | 30.08 |

members of the same group cluster together in the same area, they tend to be more cohesive. We used information provided on the Twitter user account to infer the approximate geolocation of each follower. Based on this information, we then inspected for each mosque the share of followers living in the Netherlands, and, if they were living in the Netherlands, how far away they are living from the mosque they follow on Twitter (Table 5). For geocoding of locations from Twitter profiles we used the libraries in R 'ggmaps' and 'geonames'. For calculations we used the library 'geosphere'.

We find that on average around 80% of the followers of the Twitter accounts of mosques in the Netherlands also live in the Netherlands. This appears equally so for Moroccan and Turkish mosques, Salafist and non-Salafist. When looking only at followers residing in the Netherlands, it appears that the average distance to the mosque they follow is around 40 kilometers. There is considerable variation around this mean, and 28.11% of the followers live within a 5-kilometer distance to their mosque. Those following Turkish mosques live significantly closer to their mosques compared with those following Moroccan mosques ($p < 0.001$; S5 Table). Followers of Salafist accounts on Twitter also live closer to the mosques they follow on Twitter, compared against followers of non-Salafist mosques, the difference being significant ($p < 0.001$; S5 Table). Note, however, that this difference is related to all followers; when looking at those living within 5-kilometer distance to mosques, it appears that this is more common among followers of non-Salafist mosques. Thus, based on the first measure, there is some evidence to support H3.

A second, and more direct, measure of group cohesiveness is the degree to which members of the same mosque on Twitter are connected to each other. To capture this, we performed a network analysis using the R package 'igraph' and subsequently used 'projectoR' to deal with bipartite networks. We constructed a measure of density, defined as the ratio of existing ties on Twitter between members of the same mosque to the number of all ties possible between them. We find an average density score of .22, meaning that of all possible ties between followers of the same mosque, 22% were realized (Table 6). In other words, there is 22% chance that any two randomly taken followers of a certain mosque, are also directly connected to each other on Twitter. We find that the density is rather similar across mosques, with no statistically significant differences.

However, a drawback of this density measure is that it does not capture the ties people have outside their own group. We therefore constructed another measure, which counts the number of ties people have with users who are following the same mosque as they are ("group-bonding ties"), divided by the total number of connections people have on Twitter. In this way, group-bonding ties are contrasted with the ties people have with users (pages) that are not following the same mosque as they do ("group-bridging ties"). We find that followers of Turkish and Moroccan mosques have comparable share of bonding ties (0.365 vs 0.410).

**Table 6. Group closure among mosque followers on Twitter.**

| | Bonding ties / possible bonding ties (0–1) | Bonding ties / all bonding and bridging ties (0–1) |
|---|---|---|
| All mosques | 0.221 | 0.167 |
| Ethnic group | | |
| Turkey | 0.291 | 0.365 |
| Moroccan | 0.190 | 0.409 |
| Strictness | | |
| Salafist | 0.226 | 0.380 |
| Non-Salafist | 0.220 | 0.143 |

Followers of Salafist mosques are having much higher share of bonding ties than followers of non-Salafist mosques (0.381 vs 0.143), which is in line with H3.

Another way to examine the closure of the follower network, is to consider that people can follow multiple mosques on Twitter. Using this affiliation network, we studied whether there exists a dense cluster of mosques, i.e., which have a "community structure" [68], being linked to each other by their shared followers. Fig 3 shows the unimodal projection of the network of mosques. The size of a node depicts its degree, the width of an edge shows the weight of the connection between mosques based on the number of shared followers. We differentiated mosques by ethnic group and Salafist orientation.

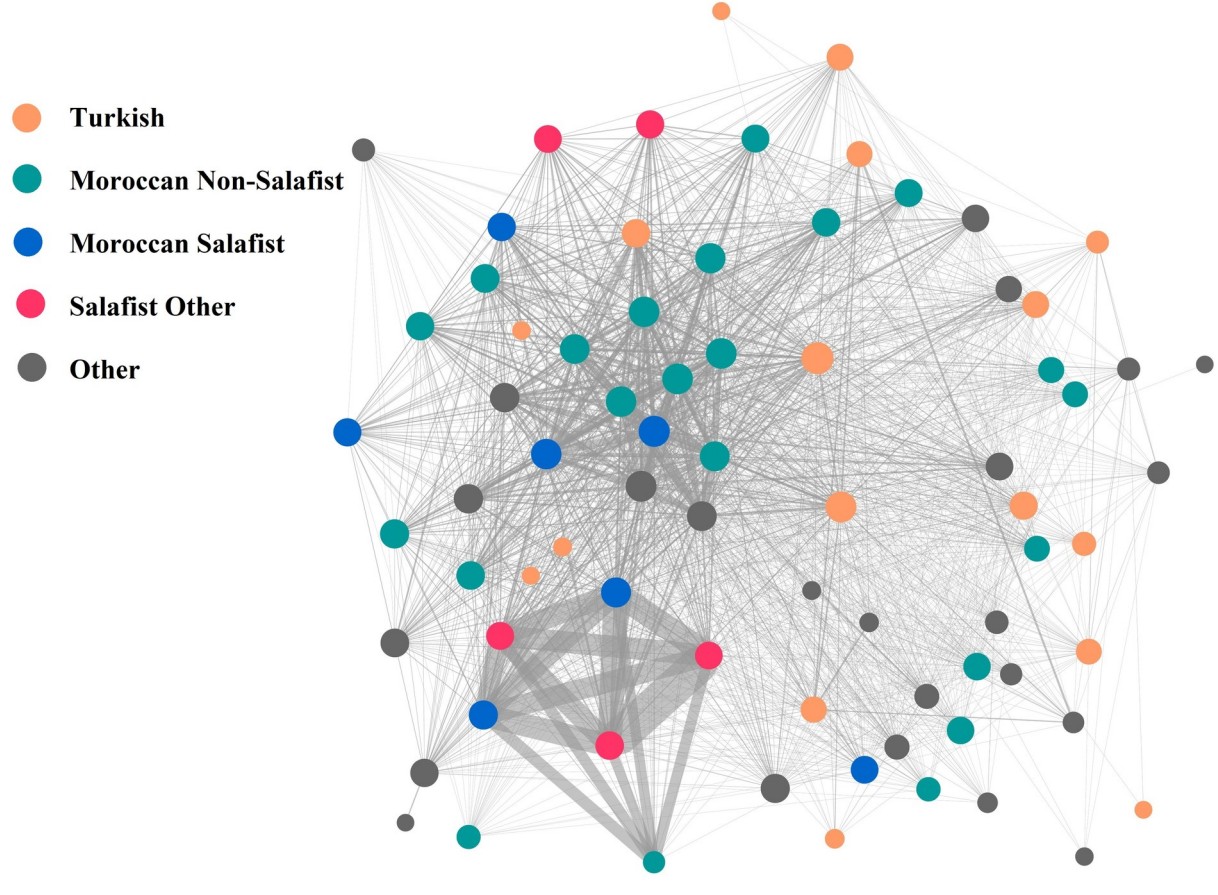

**Fig 3. Unimodal projection of the bimodal mosque-follower network.**

In order to explore the structure of the network of mosques, connected by common followers on Twitter, we used the Louvain community detection algorithm [69]. According to this algorithm, the community modularity score is 0.340. As the community modularity score can range from -1 to 1, with larger values indicating the presence of denser within-community connections and sparser connections across communities, we might conclude that the network contains communities. There appears to be three clusters in the network. The densest cluster consists of six mosques: five of those are Salafist (of which three Moroccan), and one non-Salafist. The second one comprises 46 mosques of mixed nationalities and strictness, with Moroccan non-Salafist mosques being the most prevalent group. Last, there is a cluster of Turkish mosques, comprising of 16 mosques.

To examine if there is a tendency of homophily based on ethnic group and strictness of the mosques, we calculated assortativity measures for weighted networks [70]. We used the measure of assortativity to estimate homophily, which has a range from -1 to 1, with larger values indicating stronger tendency of linkage between nodes that are similar. Results reveal no evidence for homophily based on ethnic affiliation of the mosque (assortativity = 0.0148), while we do find that there is homophily based on strictness of the mosque (assortativity = 0.3223). Hence, this means that followers of Salafist mosques tend to follow multiple Salafist mosques, creating dense clusters of interconnected Salafist mosques and followers which share similar ideology. This is again in line with H3.

## 5. Conclusion and discussion

In the past decades, the share of Muslim minority groups has strongly increased in Western Europe. Devout Muslims are exposed to a largely Christian-secular context and a substantial proportion of the ethnic majority population supports assimilationist policies. Many studies have appeared on the religiosity and integration of Muslims in Western Europe [1, 2, 7, 17]. The aim of this paper was to contribute to this literature and study whether, as a response to the secularizing-assimilationist forces in Western Europe, Muslim minority groups have built an online organizational structure that protects their Muslim identity, to examine whether mosques and their followers manifest themselves on websites, Facebook, Instagram, YouTube and Twitter. We collected data on the online activities of all mosques and their followers in the Netherlands, and relied on theories on resource mobilization and ideological strictness to derive testable hypotheses. What are the main conclusions and implications that we can draw from this study?

First, we find that mosque organizations in the Netherlands most often use websites and Facebook to manifest themselves online. Instagram, YouTube and Twitter were used considerably less often at the moment of data collection. Facebook appears to be most actively used to disseminate information, more so than other social media platforms. We also find that on Facebook, mosques attract the largest group of followers, and that followers are more actively using Facebook than other social media platforms. With an average of around 1200 followers, mosques attract roughly four times as many people as they do, on average, on Twitter, Instagram or YouTube. One implication of this finding is that, when researchers aim to further study the online presence of mosques and Muslims, they need to be aware of the strong usage of Facebook. At the same time, however, our study also suggest that some ethnic groups are more visible on certain platforms than others. Moroccan mosques, for example, are significantly less often on Facebook than Turkish mosques, but more often present on YouTube. Thus, when researchers focus on one platform only, such as Facebook, one needs to be aware of ethnic groups 'specializing' in a certain platform. Bias can arise therefore when ignoring such specialization tendencies. Our study thereby contributes to previous studies, which found

that social categories (e.g., age, gender, education) are unevenly represented on different social media platforms [31].

Second, our results suggest considerable heterogeneity among mosques and followers in their online activity and behavior. A consistent pattern we observe across platforms is that a few mosques are highly active online, and attract many (active) followers, while the majority is considerably less active and popular. This pattern (skewed distribution, long tail) resembles the well-documented power-like distributions documented on the internet and in social networks [71, 72]. Several social processes may drive this skewed distribution. Our analysis suggests that mosques that are highly active on certain social media platforms, attract more followers on that platform, but other mechanisms may also account for the skewed distribution. For example, preferential attachment may play a role [71], i.e., mosques that are followed by many people, may become more visible and more attractive to follow, leading to a positive feedback loop that generates increasing differences in online popularity of mosques over time.

Third, our more-detailed exploration of Twitter suggest that many followers live nearby the mosque they follow and are also frequently connected on Twitter to other members of the same mosque. Around 80% of the followers of a mosque in the Netherlands also live there, and of those, around 28% live within a 5-kilometer radius from the mosque. Of the people following a certain mosque, 22% are connected to each other. These findings, taken together, suggest that although anyone in the world can follow a certain mosque in the Netherlands, in reality we see high geographic clustering and density. This empirical pattern adds to a growing body of evidence, which suggests that, despite the potential to create online ties over long distances, users on social media actually tend to be strongly geographically clustered to each other [73, 74] and networks in the online world tend to mirror ties in the offline world [75, 76].

Fourth, we find no evidence to suggest that Turkish mosques are more active online than Moroccan mosques. This goes against H1, which was based on the organizational-resource argument that the Turkish group in the Netherlands has a stronger organizational structure than the Moroccan group, and that this difference would have a spillover-effect to the online world. Although Turkish mosques use Facebook and Instagram more often, and also are more active on Facebook, Moroccan mosques use YouTube more often than Turkish mosques. Overall, we do not find clear and consistent evidence for the idea that Turkish mosques are more visible and active only as compared to Moroccan mosques. Hence, stronger offline organizational cohesion and resources do not transpose into stronger online visibility and activity.

Fifth, this study provides some evidence for the role of ideology and strictness in online activity. Results suggest that the more orthodox Salafist-Muslim groups are more active online (H2), and that these groups are also a bit more cohesive (H3) than more liberal Muslim groups. Orthodox religious communities place greater penalties to free riding, screening out members who lack commitment [44]. Members of religious orthodox groups are stronger pressured to engage in pro-group activities, to recruit new members and disseminate ideology. We used this theoretical mechanism and tested the idea that Muslim groups with an orthodox ideology are more active online than more lenient Muslim groups. In line with theoretical expectations, we find that Salafist mosques significantly more often use Twitter than non-Salafist mosques, that they post more videos on YouTube, and that on Facebook they attract many more followers than non-Salafist mosques.

We also argued and indeed find that religious orthodoxy plays a role in online group closure. It was hypothesized that, due to self-imposed restrictions for interactions with out-group members, and the cultural distance with the secular Dutch majority population, the online followers of Salafist mosques cluster together. Using data on Twitter, we find that, in line with this idea, followers of the same Salafist mosque make up a dense cluster of users who are connected to each other (group-bonding ties), and much less so to other users (group-bridging

ties). Salafist mosques are furthermore strongly connected to each other, which creates a closed network of mosques that share the same followers and ideology. It therefore appears that the more-orthodox Muslim minority groups have been more successful in building an online organizational structure that responds to the secularizing-assimilationist forces in the Netherlands.

We see several limitations of this study, and suggest directions for further research to elaborate. First, we focused on mosques only. To get a more comprehensive picture of the online presence and spread of Salafist and non-Salafist Muslim ideology in Western countries, scholars are encouraged to collect data on other Muslim organizations and Muslim ideological leaders who are not representing official mosque organizations, but who may nevertheless have strong online presence and impact.

Second, our work concerned only Muslims in the Netherlands, at a single point in time. Follow-up research can bring in a more comparative and dynamic approach, such as data on the online presence and activity of mosques and followers in other countries; data on trends over time; and data about non-Muslim religious organizations.

Third, another shortcoming of this study is that only online data were used. The analysis of online data is subject to several challenges [77, 78], a key issue being the (non)representativeness of the users of social media, such as who is on Twitter, and, among those who are on Twitter, who actually tweets [31, 79], or who actually tweets with their location [80]. A promising way to continue is to link survey data with online data [81], i.e., to conduct a survey among a representative sample of Muslims and then capture their social media presence and activity.

Fourth, the aim of this study was to study the online activity and ties of mosques and their followers, and therefore remained silent on the online content disseminated. Another way to proceed would be to study ideological content of mosques and their followers on online platforms, taking advantage of the rise of large-scale, automated text analysis [82, 83]. It would be particularly relevant to examine the dynamics between, on the one hand the anti-Muslim, anti-Islam frames and rhetoric expressed in the news and on social media [84, 85], and the response to these sentiments in Muslim communities on the other hand.

## Supporting information

**S1 Fig. Distribution on platforms, user activity.**
(TIF)

**S1 Text. Results procedural note.**
(DOCX)

**S2 Text. Data collection.**
(DOCX)

**S1 Table. Regression models of online presence.**
(DOCX)

**S2 Table. Regression models of mosque activity.**
(DOCX)

**S3 Table. Regression models of number of followers.**
(DOCX)

**S4 Table. Regression models of user activity.**
(DOCX)

**S5 Table. Regression models of geodistance.**
(DOCX)

## Author Contributions

**Conceptualization:** Frank van Tubergen.

**Data curation:** Frank van Tubergen, Tobias Cinjee, Anastasia Menshikova, Joran Veldkamp.

**Formal analysis:** Anastasia Menshikova, Joran Veldkamp.

**Investigation:** Frank van Tubergen, Tobias Cinjee, Anastasia Menshikova, Joran Veldkamp.

**Methodology:** Frank van Tubergen, Tobias Cinjee, Anastasia Menshikova.

**Supervision:** Frank van Tubergen.

**Writing – original draft:** Frank van Tubergen.

**Writing – review & editing:** Frank van Tubergen, Tobias Cinjee.

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
