## [Decision Letter · Decision Letter 0]

2 Nov 2020

PONE-D-20-19582

Online activity of mosques and Muslims in the Netherlands: a study of Facebook, Instagram, YouTube and Twitter

PLOS ONE

Dear Dr. van Tubergen,

Thank you for submitting your manuscript to PLOS ONE. After careful consideration, we feel that it has merit but does not fully meet PLOS ONE’s publication criteria as it currently stands. Therefore, we invite you to submit a revised version of the manuscript that addresses the points raised during the review process.

I am giving a major revision though was inclined to reject the paper because two of the reviewers have given a reject and the third reviewer while giving a minor revision have not commented/evaluated the manuscript in terms of the key characteristics required by an article of a journal of PLOS ONE standing. Please see the journal publication criteria.

The quality issues firstly relate to the clarity of the aims and objectives of the article. What is the aim here and what is its scientific and social impact. Please also address the ethical concerns of the study. Moreover, the reviewers have questioned the methodology and results. Please carefully address the reviewers' comments in preparing the revised version and provide a point-wise response to each of the reviewers concerns. 

We look forward to receiving your revised manuscript.

Kind regards,

Rashid Mehmood, PhD

Academic Editor

PLOS ONE

Journal Requirements:

Reviewers' comments:

Reviewer's Responses to Questions

**Comments to the Author**

1. Is the manuscript technically sound, and do the data support the conclusions?

Reviewer #1: Yes

Reviewer #2: No

Reviewer #3: Partly

2. Has the statistical analysis been performed appropriately and rigorously? 

Reviewer #1: Yes

Reviewer #2: No

Reviewer #3: Yes

3. Have the authors made all data underlying the findings in their manuscript fully available?

Reviewer #1: Yes

Reviewer #2: Yes

Reviewer #3: Yes

4. Is the manuscript presented in an intelligible fashion and written in standard English?

Reviewer #1: Yes

Reviewer #2: No

Reviewer #3: Yes

5. Review Comments to the Author

Reviewer #1: The authors present a scientific study into the use of online forums by mosques in Netherlands. The paper is well-written and structured in a way that makes it easy to follow. Specific comments are:

1. Average distance is currently calculated country-wide but perhaps can be refined further into regions or cities etc to identify which mosques have followers from other cities indicating hubs…

2. have the ownership of social media pages certified? It may be the case that a facebook page is not owned by the mosque...

3. authors indicate that youtube is not used frequently by the mosques but Facebook is. Perhaps further analysis can be conducted in terms of type of content? So it will be interesting to see the type of content published on Facebook and other forums…

4. Some figures require improvement with respect to their quality.

Reviewer #2: In this paper, the authors analyze the online activities of muslims on social media. It is unclear what is the scientific contribution in the paper. A review and comparison of the related works are missing. What is the gap in the current literature?

What is the scientific value and novelty of the proposed solution? What is the relevance of this paper to the journal?

What is the analysis of online activities of different minority communities on social media? How to avoid any bias in such an analysis? How the results from such analysis can be used to ensure better facilities, social justice, equality. How to avoid any ethical issues, or avoid bias, stigmatization on minorities [1].

[1] S. Loue, "Ethical Issues in Conducting Research on Religion and Spirituality",

How can such analysis be used to measure equality, diversity, social justice for minorities? What is the purpose of this analysis? What methods and algorithms can be used to carry out such analyses? There are a lot of open-ended questions that may arise which should be answered. This paper should be thoroughly revised and should be re-submitted.

Reviewer #3: This is an interesting manuscript that explores the online behavior of mosques and participants across the Netherlands.

While there is much to appreciate in this manuscript, it does not warrant publication in PLOS ONE.

First, the piece is very descriptive and lacks a positivist research focus. This is quite problematic for an article of this nature. Oftentimes descriptive pieces are either uncovering empirically novel insights or data collection strategies that are original. Unfortunately, neither applies here.

Second, the lack of theoretical engagement is problematic, because it is unclear what to take away from the different research questions. Why should we care about the differences between Turkish and Moroccan mosques? What do theories about immigrant assimilation/acculturation suggest? Or is this about Islamic brand/theology?

Third, why should we be surprised that Salafist mosques have a significant online presence? Most right-wing/fundamental/radical groups tend to exhibit similar trends.

Finally, I am not entirely convinced that mosque “participants” are captured adequately. The ms takes twitter followers as a measure of participation. This is a major unsubstantiated assumption. Many follow mosques, even though they aren’t participants. So I wonder about the validity of this measure. There should have been an effort to substantiate this claim.

The major frame of the paper is that online analyses can uncover new phenomena, insights and patterns that conventional political behavior measures miss. Unfortunately, the ms does not accomplish this goal.

6. PLOS authors have the option to publish the peer review history of their article (what does this mean?). If published, this will include your full peer review and any attached files.

Reviewer #1: No

Reviewer #2: No

Reviewer #3: No

---

## [Author Response · Author response to Decision Letter 0]

18 Jan 2021

We would like to thank the reviewers for taking the time to read our paper and to give feedback. It really helped us to improve the paper. 

Reviewer #1: 

1. Average distance is currently calculated country-wide but perhaps can be refined further into regions or cities etc to identify which mosques have followers from other cities indicating hubs.

This is an interesting idea. In the paper, we look at the geographic distance between the mosques in the Netherlands and the place of living of their ‘followers’ on Twitter. We provide different pieces of information. First, we show how many of the followers of the mosques actually live in the Netherlands or abroad. This gives us a first idea of how nationally or internationally embedded followers of mosques are that are located in the Netherlands. Then, we take a closer look at those mosque-followers who live in the Netherlands, providing two further pieces of information: (a) what is their average distance to the mosque, (b) how many live very nearby (less than 5 kilometer from the mosque). We like the suggestion to go beyond this analysis and also look at the cities from which mosques attract followers. Unfortunately, this raises an issue of ethics and privacy, because revealing the geographical location of mosques would not be in line with GDPR rules. The data collection for the project has been approved by the ethics committee of the university, but this approval does not include revealing geographical data which could lead to identification of mosques. Therefore, we cannot provide information of the kind that “the mosque located in city X appear to be a major hub in the Netherlands,” or “city X attracts followers from Y”, because if there is only one such mosque in city X, and that happens, we effectively reveal the identity of the mosque. That’s why we present results on the share of mosque-followers living nearby. 

2. Have the ownership of social media pages certified? It may be the case that a Facebook page is not owned by the mosque.

This is a good point. We checked the social media pages extensively, one by one, and included pages on mosques in the Netherlands that are created by (a) an organizational account of the mosque, or (b) a person on behalf of the mosque. To qualify for (b), (a) should not be present on the same social media platform, and, moreover, it should be clear from the content and number of followers that it is in fact the main social media account (e.g., on Facebook) that is used to spread information about the mosque. We clarify this in the Revision. 

3. Authors indicate that YouTube is not used frequently by the mosques but Facebook is. Perhaps further analysis can be conducted in terms of type of content? So it will be interesting to see the type of content published on Facebook and other forums.

This is an interesting suggestion. We looked at many Facebook pages of mosques, and what we discovered is that actually a lot of visual, non-textual information was provided. Pictures and videos of meetings, lectures, gatherings and social events. The content of these visual data complicates the analysis, because the amount of visual and textual data strongly varies from mosque to mosque (e.g., one mosque posting mainly pictures and videos, while another mosque posts text). Such a study, however interesting, goes beyond the scope of this paper. It is for these reasons that we have refrained from such an analysis in this paper, but we do mention this idea as an interesting step forward for follow-up research. 

4. Some figures require improvement with respect to their quality.

We checked all figures, and discovered one figure (Fig 3) which was of lower quality (144dpi instead of 300dpi). We corrected this.

Reviewer #2: 

1. A review and comparison of the related works are missing. What is the gap in the current literature? What is the scientific value and novelty of the proposed solution? What is the relevance of this paper to the journal?

We agree that the contributions of this study could be better articulated. We have completely rewritten the Introduction, included a thorough review of previous work, and clearly outlined how this paper significantly contributes to the state-of-the art. 

2. What is the analysis of online activities of different minority communities on social media? How to avoid any bias in such an analysis? How the results from such analysis can be used to ensure better facilities, social justice, equality. How to avoid any ethical issues, or avoid bias, stigmatization on minorities [1]. [1] S. Loue, "Ethical Issues in Conducting Research on Religion and Spirituality",

We read the paper suggested by the reviewer, and what it says is in line with GDPR guidelines. To clarify, our paper has been reviewed and approved by the University Ethics Committee, which means that our work presented here is 100% following GDPR rules.

3. How can such analysis be used to measure equality, diversity, social justice for minorities? What is the purpose of this analysis? What methods and algorithms can be used to carry out such analyses? There are a lot of open-ended questions that may arise which should be answered. This paper should be thoroughly revised and should be re-submitted.

We’re not sure if we understand the points the reviewer wants to make. The aims of the paper are not to measure equality, diversity or social justice for minorities. We clearly formulate the aims and background of the study, research questions, contributions, theory and hypotheses, methods, results and implications. We also submitted Supplemental materials, which contain more detailed information on the methods and algorithms that we used. 

Reviewer #3: 

1. First, the piece is very descriptive and lacks a positivist research focus. This is quite problematic for an article of this nature. Oftentimes descriptive pieces are either uncovering empirically novel insights or data collection strategies that are original. Unfortunately, neither applies here.

We strongly embrace a theory-driven, analytical (‘positivist’) research approach. The questions we address in this study are not only descriptive. On the contrary, we also raise explanatory questions, we discuss several theoretical mechanisms that help to understand differences between Muslim groups in their adoption of online platforms and derive several hypotheses which are then tested in the quantitative analyses. We suspect that we have not emphasized this theoretical part very well in the previous version of this paper, so it may have indeed appeared as if we only introduced descriptive material. In this Revision, we have emphasized more strongly the theoretical contributions of the paper and included an entire section on Theory and Hypotheses. Note that with respect to the descriptive analyses, we’re the first to present findings from a large-scale comparative study on Muslims adoption of online platforms. This is a significant contribution to most work in this field, which takes a more qualitative, case-study approach (from which it’s hard to make inferences to other groups and platforms). 

2. Second, the lack of theoretical engagement is problematic, because it is unclear what to take away from the different research questions. Why should we care about the differences between Turkish and Moroccan mosques? What do theories about immigrant assimilation/acculturation suggest? Or is this about Islamic brand/theology?

We agree that in the previous version, these theoretical questions were not articulated very well. We also agree that the discussion of previous work and theories could be more elaborate. We have therefore entirely rewritten the Introduction, incorporating a much strong review of previous work. We have also clearly formulated the three contributions of this paper and included a separate section on Theory and Hypotheses. 

3. Third, why should we be surprised that Salafist mosques have a significant online presence? Most right-wing/fundamental/radical groups tend to exhibit similar trends.

In this paper, we look not only at the online presence and activity of mosques, but also at their online ‘followers’. A key question regarding the status of Muslim minority groups in Western Europe, is how they respond to the secular and assimilationist forces. What we find is that not only are the orthodox Salafist mosques more active online, but also that they have more online followers, and that these followers make up a more cohesive group: they cluster together in the same areas, and online they make up a cohesive web of in-group ties. This study is the first to show that the most-orthodox Muslim groups have been more successful than other Muslim groups in building an online organizational structure that responds to the secularizing-assimilationist forces in the Netherlands. 

4. Finally, I am not entirely convinced that mosque “participants” are captured adequately. The ms takes twitter followers as a measure of participation. This is a major unsubstantiated assumption. Many follow mosques, even though they aren’t participants. So I wonder about the validity of this measure. There should have been an effort to substantiate this claim.

This is a very good point. We agree that a ‘follower’ of a mosque online does not imply that s/he also participates in meetings of that mosque offline. Everyone can follow mosques online, and thus a ‘follower’ of a Turkish mosque on Twitter does not mean that this follower attends meetings of that mosque. We clarify this issue in the paper, note that this possible ‘bias’ applies to all mosques we study (we have no evidence to suggest it affects some mosques more than other mosques), and ceteris paribus test the hypotheses.

---

## [Decision Letter · Decision Letter 1]

9 May 2021

PONE-D-20-19582R1

Online activity of mosques and Muslims in the Netherlands: a study of Facebook, Instagram, YouTube and Twitter

PLOS ONE

Dear Dr. van Tubergen,

Thank you for submitting your manuscript to PLOS ONE. After careful consideration, we feel that it has merit but does not fully meet PLOS ONE’s publication criteria as it currently stands. Therefore, we invite you to submit a revised version of the manuscript that addresses the points raised during the review process.

Please address the reviewer's comments and address the PLOS ONE’s publication criteria while revising your manuscript. 

We look forward to receiving your revised manuscript.

Kind regards,

Rashid Mehmood, PhD

Academic Editor

PLOS ONE

Reviewers' comments:

Reviewer's Responses to Questions

**Comments to the Author**

1. If the authors have adequately addressed your comments raised in a previous round of review and you feel that this manuscript is now acceptable for publication, you may indicate that here to bypass the “Comments to the Author” section, enter your conflict of interest statement in the “Confidential to Editor” section, and submit your "Accept" recommendation.

Reviewer #3: (No Response)

2. Is the manuscript technically sound, and do the data support the conclusions?

Reviewer #3: Partly

3. Has the statistical analysis been performed appropriately and rigorously? 

Reviewer #3: (No Response)

4. Have the authors made all data underlying the findings in their manuscript fully available?

Reviewer #3: (No Response)

5. Is the manuscript presented in an intelligible fashion and written in standard English?

Reviewer #3: Yes

6. Review Comments to the Author

Reviewer #3: I still find the theoretical justification/contribution weak and lacking coherency.

The existing rationale for the comparison of Moroccan and Turkish mosques also doesn’t work particularly well in this paper.

One suggestion is to further build the theory being tested. It appears that the Turkish mosques have better resources than the Moroccan ones as the author stipulates. One way to motivate the study is to ask whether organizational resources or ideology serve as better predictors of online networks. This would be directly engaging theories of resource mobilization and the role of affluence in acculturation/assimilation models against ideological commitments which may structure ingroup linkages/biases.

I believe that Moroccans are a more recent community (?) So that can be juxtaposed against the Turks who have probably been in the country for a longer period of time. If this is correct, and there are no differences between the two communities, it illustrates the time in host country is less consequential to online networking.

To examine the resource-based argument, the author can proceed to compare Moroccan to Turkish mosques. To examine the ideological explanation, the author will look at Salafist/non-Salafist mosques as is already done in the paper. (Please note that Salafists only constitute 5% of mosques, so this should be highlighted. Further, the author should provide the coding scheme for Salafist vs non-Salafist mosques.) The author can also look at the role of time in host country, as yet a third dimension of theoretical interest.

Other Points smaller:

On page 11, there’s a discussion of Pakistani mosques, but it’s unclear whether these are more comparable to Turkish or Moroccan mosques?

Page 18: 5-kilometer distance to mosque (%)-----it appears a larger percentage of non-Salafists live in this radius. So I would relax the language here about Salafists living closer to mosques.

7. PLOS authors have the option to publish the peer review history of their article (what does this mean?). If published, this will include your full peer review and any attached files.

Reviewer #3: No

---

## [Author Response · Author response to Decision Letter 1]

25 May 2021

Response to Reviewers

We would like to thank Reviewer #3 once more for taking the time to read our paper and to give feedback. 

1. One suggestion is to further build the theory being tested. It appears that the Turkish mosques have better resources than the Moroccan ones as the author stipulates. One way to motivate the study is to ask whether organizational resources or ideology serve as better predictors of online networks. This would be directly engaging theories of resource mobilization and the role of affluence in acculturation/assimilation models against ideological commitments which may structure ingroup linkages/biases.

This is a good suggestion. We followed your advice and included already in the introduction this theoretical contrast between organizational resources and ideology as drivers for online activity. 

2. I believe that Moroccans are a more recent community (?) So that can be juxtaposed against the Turks who have probably been in the country for a longer period of time. If this is correct, and there are no differences between the two communities, it illustrates the time in host country is less consequential to online networking.

Turks and Moroccans arrived at the same time, as part of the ‘guest-worker’ program of the Dutch government in the 1960s. This is now clarified in the introduction. 

3. To examine the resource-based argument, the author can proceed to compare Moroccan to Turkish mosques. To examine the ideological explanation, the author will look at Salafist/non-Salafist mosques as is already done in the paper. 

This is what we now do in the paper, see Hypothesis 1 (resource argument) and Hypothesis 2 and 3 (ideology). 

4. (Please note that Salafists only constitute 5% of mosques, so this should be highlighted. Further, the author should provide the coding scheme for Salafist vs non-Salafist mosques.) 

We now highlight the 5% in the revised manuscript. Regarding coding of Salafist/non-Salafist mosques, we rely on previous work and include references to these studies in the current paper. We didn’t do the coding ourselves, therefore, but instead relied on previous work that coded mosques as Salafist based on a variety of sources, such as Salafist self-identification of mosques, anthropological fieldwork, and information on the ideology of mosques. We were able to match the names of these mosques mentioned in these studies to our list of 478 mosques. 

5. The author can also look at the role of time in host country, as yet a third dimension of theoretical interest.

This would be interesting to study, but because the groups with enough mosques for analysis (Turks, Moroccans) arrived at the same time, we cannot study it. 

6. Other Points smaller: On page 11, there’s a discussion of Pakistani mosques, but it’s unclear whether these are more comparable to Turkish or Moroccan mosques?

That’s a difficult question, we know very little about Pakistani, their organizational structure and ideology -it’s a very, very small group in the Netherlands (< 25,000 immigrants and children of immigrants are from Pakistani origin). It’s remarkable that they are so-well organized online, that’s why we mentioned it in the descriptive part of the paper.

7. Page 18: 5-kilometer distance to mosque (%)-----it appears a larger percentage of non-Salafists live in this radius. So I would relax the language here about Salafists living closer to mosques.

We have changed the text accordingly.

---

## [Decision Letter · Decision Letter 2]

7 Jul 2021

Online activity of mosques and Muslims in the Netherlands: a study of Facebook, Instagram, YouTube and Twitter

PONE-D-20-19582R2

Dear Dr. van Tubergen,

We’re pleased to inform you that your manuscript has been judged scientifically suitable for publication and will be formally accepted for publication once it meets all outstanding technical requirements.

Kind regards,

Rashid Mehmood, PhD

Academic Editor

PLOS ONE

Additional Editor Comments (optional):

Reviewers' comments:

Reviewer's Responses to Questions

**Comments to the Author**

1. If the authors have adequately addressed your comments raised in a previous round of review and you feel that this manuscript is now acceptable for publication, you may indicate that here to bypass the “Comments to the Author” section, enter your conflict of interest statement in the “Confidential to Editor” section, and submit your "Accept" recommendation.

Reviewer #3: All comments have been addressed

2. Is the manuscript technically sound, and do the data support the conclusions?

Reviewer #3: (No Response)

3. Has the statistical analysis been performed appropriately and rigorously? 

Reviewer #3: (No Response)

4. Have the authors made all data underlying the findings in their manuscript fully available?

Reviewer #3: (No Response)

5. Is the manuscript presented in an intelligible fashion and written in standard English?

Reviewer #3: (No Response)

6. Review Comments to the Author

Reviewer #3: (No Response)

7. PLOS authors have the option to publish the peer review history of their article (what does this mean?). If published, this will include your full peer review and any attached files.

Reviewer #3: No

---

## [Editor Report · Acceptance letter]

13 Jul 2021

PONE-D-20-19582R2 

Online activity of mosques and Muslims in the Netherlands: a study of Facebook, Instagram, YouTube and Twitter 

Dear Dr. van Tubergen:

I'm pleased to inform you that your manuscript has been deemed suitable for publication in PLOS ONE. Congratulations! Your manuscript is now with our production department. 

Kind regards, 

on behalf of

Dr. Rashid Mehmood 

Academic Editor

PLOS ONE